# New Technology of Rumen-Protected Bypass Lysine Encapsulated in Lipid Matrix of Beeswax and Carnauba Wax and Natural Tannin Blended for Ruminant Diets

**DOI:** 10.3390/ani14192895

**Published:** 2024-10-08

**Authors:** Claudiney Felipe Almeida Inô, José Morais Pereira Filho, Roberto Matheus Tavares de Oliveira, Juliana Felipe Paula de Oliveira, Edson Cavalcanti da Silva Filho, Ariane Maria da Silva Santos Nascimento, Ronaldo Lopes Oliveira, Romilda Rodrigues do Nascimento, Kevily Henrique de Oliveira Soares de Lucena, Leilson Rocha Bezerra

**Affiliations:** 1Graduate Program in Animal Science and Health Animal Science Department, Federal University of Campina Grande, Patos 58708-110, Paraíba, Brazil; claudiney.felipe@estudante.ufcg.edu.br (C.F.A.I.); jmorais@cstr.ufcg.edu.br (J.M.P.F.); r.matheustavares98@gmail.com (R.M.T.d.O.); romildarn01@ufpi.edu.br (R.R.d.N.); hkevily@gmail.com (K.H.d.O.S.d.L.); 2Campus do Sertão, Federal University of Sergipe, Nossa Nova Esperança, Nossa Senhora da Glória 49680-000, Sergipe, Brazil; jupaula.oliv@academico.ufs.br; 3LIMAV, Interdisciplinary Laboratory for Advanced Materials, Campus Ministro Petrônio Portella, Piaui Federal University, Teresina 64049-550, Piaui, Brazil; edsonfilho@ufpi.edu.br (E.C.d.S.F.); ariane.am42@gmail.com (A.M.d.S.S.N.); 4Animal Science Department, Federal University of Bahia, Salvador 40170-155, Bahia, Brazil; ronaldooliveira@ufba.br

**Keywords:** amino acid, efficiency, emulsification, microencapsulation, protection, ruminant, sheep

## Abstract

**Simple Summary:**

Using bypass protein in ruminants’ diets is crucial as it allows essential amino acids, like lysine, to escape rumen degradation and be absorbed in the intestine. This improves protein utilization and enhances growth, milk production, and overall animal health, leading to more efficient and sustainable livestock farming. This study explores the use of beeswax (BW) and carnauba wax (CW) lipid matrices to create bypass lysine (Lys) for ruminants, with tannins from *Mimosa tenuiflora* hay enhancing the protection of Lys. The research tested eight microencapsulated treatments with varying tannin levels (0%, 1%, 2%; 3%) using the fusion–emulsification technique. The results showed that adding tannins improved the microencapsulation yield and efficiency. Among the treatments, CWLys_3%_ demonstrated the highest efficiency in retaining Lys and provided superior protection against rumen degradation. Our study demonstrated the efficiency of encapsulating the material (bypass protein) and making it available for the nutrition and growth of sheep. Future studies should determine what levels can be added to the diet and the consequences of using Lys bypass on the digestion and metabolism of sheep.

**Abstract:**

Tannins are compounds present in forage plants that, in small quantities in the diet of ruminants, produce protein complexes that promote passage through the rumen and use in the intestine. This study tested the hypothesis that beeswax (BW) and carnauba wax (CW) lipid matrices are effective encapsulants for creating bypass lysine (Lys) for ruminants, with tannin extracted from the *Mimosa tenuiflora* hay source enhancing material protection. Microencapsulated systems were made using the fusion–emulsification technique with a 2:1 shell-to-core ratio and four tannin levels (0%, 1%, 2%; 3%). The following eight treatments were tested: BWLys_0%_, BWLys_1%_, BWLys_2%_, BWLys_3%_, CWLys_0%_, CWLys_1%_, CWLys_2%_, and CWLys_3%_. Tannin inclusion improved microencapsulation yield and efficiency. CWLys_3%_ had the highest microencapsulation efficiency and retained Lys. Lysine in BW and CW matrices showed higher thermal stability than in its free form. Material retention was greater in BW than CW. Rumen pH and temperature remained unaffected, indicating that BW and CW as the shell and tannin as the adjuvant are efficient encapsulants for Lys bypass production. The formulation CWLys_3%_ is recommended as it is more efficient in protecting the lysin amino acid from rumen degradation.

## 1. Introduction

The protein requirements of ruminant animals, met by the amino acids absorbed in the small intestine, are called metabolizable protein requirements [1,2]. Microbial protein synthesized in the rumen is a crucial source of protein for ruminant animals, providing 50% or more of the amino acids (AAs) available for absorption in cattle diets. Microbial crude protein (MCP) has a high-quality and relatively consistent amino acid profile, which makes it challenging to alter the amino acid composition in the duodenal digesta. Since MCP accounts for more than 50% of the protein digested by ruminants, understanding its amino acid composition is essential for accurately estimating the overall AA supply [3,4,5].

However, ruminant animals also require adequate amounts of essential post-rumen amino acids to meet their maintenance and production needs. In addition, knowledge about the nutritional requirements of these amino acids, as well as the transformations that the feed undergoes during rumen fermentation, make it difficult to determine the amino acids available for absorption in the duodenum, which come from a mixture of microbial, dietary overpass and endogenous protein [6,7].

Thus, the protection of amino acids for passage through the rumen and reaching the intestine needs to be considered since diets containing different protein sources result in changes in the amount and pattern of amino acids due to the conversion of microbial protein in the rumen [8]. This phenomenon will be very advantageous, as depending on the quality of the protein ingested, it will be possible to provide greater amino acid passages and ensure that the animals’ post-rumen requirements are met. In addition, high-quality proteins can have their nutritional value reduced by rumen fermentation, while low-quality proteins can be transformed into proteins of high biological value [9].

There is great interest in determining amino acid requirements and developing systems that allow diets to be balanced based on the amino acids absorbed in the intestine [5]. One of the reasons is that the amino acid profile of the protein not degraded in the rumen may not be adequate to maximize the use of metabolizable protein in protein synthesis by the animal [10]. Therefore, protecting part of the protein in the diet from rumen degradation is an excellent nutritional alternative, especially when associated with non-protein nitrogen (NNP) sources, maximizing the production of metabolizable protein, which is the protein fraction that matters. This strategy is very interesting when aspects of feed efficiency and maximizing results are sought.

Microencapsulation is a process that consists of coating a substance (core) with a thin layer of another material (wall), forming microscopic particles called microcapsules that can be used to protect substances sensitive to degradation [11,12,13]. The microencapsulation technique can protect amino acids in the diet of ruminants and allows controlled release at specific sites in the gastrointestinal tract [14]. To allow efficient transport of the nucleus, methionine, or lysine (Lys) amino acids, the protective substance must protect the compound from attacks by the microbiota in the animal’s rumen and gradually release this active substance into the small intestine [15].

Therefore, several studies point to the need to protect amino acids from rumen degradation [6,16,17,18] by making them more accessible, maximizing their absorption in the small intestine and improving animal productivity. Thus, we hypothesized that carnauba (CW) and beeswax (BW) can provide efficient protection of Lys against rumen degradation and that this protection can be enhanced with the use of condensed tannin as an adjuvant. This is because, in general and depending on the quantity, when added to ruminant diets, tannins form complexes with proteins, as well as being insoluble and stable under rumen conditions, while at the same time modulating rumen fermentation [19]. At low and moderate concentrations, tannins reduce protein degradation in the rumen, with dissociation only occurring in the abomasum, which is a factor that defines better utilization of dietary protein, unlike what occurs in the rumen environment [20].

Tannins have positive effects related to their beneficial properties as antioxidants, radical scavengers and modulators of the intestinal microbiota [21,22]. These properties make this product an interesting adjuvant to complement animal diets. However, the same chemical mechanisms that give them some of their positive effects (binding proteins) are also responsible for their astringent taste [23]. On the other hand, the interaction of tannins with proteins and carbohydrates could also determine the effects on the encapsulating matrix to which they are added [24].

The purpose of our study was to develop and apply a new encapsulated microstructure of the bypass protein based on Lys as the core using the emulsification method and the lipid matrix of carnauba and beeswax as the shell, enriching it with natural tannins, characterizing the material and evaluating its inclusion in ruminant diets.

## 2. Materials and Methods

### 2.1. Ethical Considerations

The experiment was carried out after approval by the Ethics Committee on Animal Experiments (CEUA) of the Federal University of Campina Grande (UFCG) under Protocol CEUA/UFCG numbers 38/202 and 39/2022.

### 2.2. Collection, Preparation and Extraction of Tannin from Mimosa tenuiflora Hay

The natural tannin was obtained from hay of *Mimosa tenuiflora*, popularly called Jurema Preta. The *Mimosa tenuiflora* was collected in areas of native pasture of Patos city, Paraíba, Brazil, located in the semi-arid region at coordinates 07°04′49.68″ south latitude and 037°16′22.85″ west longitude, and an altitude of 264 m, with an average temperature of 29 °C, 50.5% air relative humidity, and average rainfall of 500 mm. 

The haying was realized from the leaves and branches up to 8.0 mm in diameter from plants at full vegetation stage and an average height of 3.0 m. The material collected was shredded in a chipper and dried on plastic tarpaulins in the open air, turning it over every two hours and covering it overnight until it reached the hay stage. After the material was completely dry, it was crushed in a forage chopper machine using a sieve with diameter of 5.0 mm. 

Tannin was extracted using the modified methodology proposed by Chaves [25], in which 15 L of distilled water was used for every 1.0 kg of hay. The mixture (water + hay) was placed in a 30 L stainless steel container, and then subjected to the boiling temperature of water and ambient pressure in an autoclave with a capacity of 48 L. Extraction occurred for two hours, with the temperature ranging from 100 °C to 110 °C, avoiding degradation due to oxidation of the tannins and to obtain the maximum amount of tannic extract. At each stage, the material was filtered through porous fabric and the extract obtained was filtered through a sieve with silkscreen fabric. The filtered material was then placed in glass trays (5 × 40 × 60 cm) and subjected to natural evaporation until 50% of the liquid was lost in a forced air circulation oven at 100 °C, until reaching constant weight. The materials were then ground in an industrial blender and homogenized using a sieve for subsequent enrichment in the encapsulating material (waxes).

### 2.3. Preparation of Microencapsulated Systems to Protect Lysine (Experimental Treatments)

After obtaining the tannin extract, a completely randomized experiment was designed in a 2 × 4 factorial arrangement for lysine protection, with two types of encapsulants (shell), beeswax (BW) and carnauba wax (CW) for lysine (Lys) protection (core), with a shell-to-core ratio of 2:1 and four levels of tannin, i.e., 0; 1; 2; 3% inclusion based on the dry matter of the wax mass as the adjuvant to the material, totaling eight experimental treatments, including BWLys_0%_; BWLys_1%_; BWLys_2%_; and BWLys_3%_ and CWLys_0%_; CWLys_1%_; CWLys_2%_; and CWLys_3%_.

The preparation of microencapsulated Lys as a bypass protein source was carried out according to Carvalho Neto et al. [14] with adaptations for BW and Lys. Tannin was added according to the adsorption capacity of the encapsulating material. The tannin from *Mimosa tenuiflora* was weighed on an analytical scale, placed in beakers and dissolved in distilled water. Beeswax, CW and soy lecithin (5% based on the mass of the wax) were weighed in another beaker and melted in a water bath at 65 °C and 85 °C, respectively. The Lys was heated to the same temperature as the molten wax and then added slowly to the waxes, followed by constant stirring with a stick for 20 min for BW and 10 min for CW. The material was dried in a forced air circulation oven at 55 °C for 6 h, obtaining the microencapsulated products (treatments), which were crushed and packed in a suitable container and kept at room temperature.

### 2.4. Evaluation and Characterization of the Microencapsulated Systems

The microencapsulation yield was based on the masses of Lys, wax with and without tannin and soy lecithin (emulsifier) and the mass after drying (final), using Equation (1), according to Medeiros et al. [11].
MY = (FM/IM) × 100,(1)
where MY is the microencapsulation yield; FM is the mass of the microencapsulated product after drying; IM is the dry mass of Lys, tannin, wax and lecithin. The efficiency of the Lys microencapsulation process using the wax matrix was determined by the ratio between the retained and inserted Lys content using Equation (2) according to Medeiros et al. [11].
%ME = (L_retained_/L_inserted_) × 100,(2)
where ME is the microencapsulation efficiency; Lretained is the actual retained Lys content; L_inserted_ is the inserted Lys content.

The differential scanning calorimetry curves were obtained using the differential scanning calorimeter model Differential Scanning Calorimetry −60 from the brand Shimadzu (Tokyo, Japan) under an inert atmosphere (nitrogen) in a flow of 50 mL/min at a heating rate of 10 °C/min and temperature range of 30 to 400 °C, using a platinum crucible containing around 3 mg of the sample. The differential scanning calorimetry curves were plotted, and the data were analyzed using OriginPro 8 software, considering the peak temperature of the events.

Water activity was assessed using a model water activity meter (LabStart-aw-Novasina, Washington, DC 20037, USA). To take the micrographs, the samples were fixed to an aluminum substrate (stub) using double-sided adhesive tape and then metallized with gold. The micrographs were taken using an FEI model Quanta FEG 250 (FEI company, Hillsboro, OR, USA) field emission scanning electron microscope, with an acceleration voltage of 15 kV, spot size 5.0 and a vCD detector (low voltage, high contrast detector), in accordance with the recommendations of Dedavid et al. [26].

### 2.5. In Vitro Degradability (DaisyII Ankom)

The method known as the DaisyII rumen fermenter (ANKOM Technology Corp., Fairport, NY, USA) was used to assess in vitro degradability. Samples of Lys and microencapsulated waxes associated with different levels of tannic extract were subjected to in vitro release analysis of dry matter (DM) and crude protein (CP), using the technique described by ANKOM Technology [27]. 

The rumen liquid used was collected from a slaughterhouse in the town of Patos, Paraíba, which was filtered and stored in a thermal container and taken to LANA, with 300 mL of the rumen liquid being transferred to each of the jars of the DaisyII ANKOM rumen fermenter. The pH was kept at 6.8 to incubate the bags. Non-woven fabric (TNT) bags measuring 5 cm × 5 cm (width and length), with a pore size of 50 μm, were used and sealed with a heater.

Each jar had a plastic lid with a one-way valve to prevent the accumulation of fermentation gases. The bags were weighed and then 2.0 g of the formulations (BWLys_0%_; BWLys_1%_; BWLys_2%_; and BWLys_3%_ and CWLys_0%_; CWLys_1%_; CWLys_2%_; and CWLys_3%_) were added. The incubation times used were 0, 30, 60, 120, 180, 1440 and 2880 min, which were kept in the rotary incubator (DaisyII ANKOM, Macedon, New York, NY, USA) at 39 °C, with continuous rotation to facilitate effective immersion of the bags in the liquid portion of the jar. Carbon dioxide gas (CO_2_) was added to remove the bags at the initial times to maintain life in the rumen inoculum. After incubation, the bags were washed in cold water and dried in an oven at 55 °C for 72 h. The ingredients used in each jar to incubate the bags were sodium bicarbonate (15.2 g), sodium phosphate (5.8 g), potassium chloride (0.88 g), sodium chloride (0.73 g), magnesium sulphate (0.19 g), calcium chloride (0.08 g), urea (26 mL), glucose (26 mL), distilled water (1.54 L) and rumen fluid (300 mL).

### 2.6. In Situ Rumen Degradability Trial

Three rumen-fistulated sheep weighing around 40 kg were used for the in situ degradability test. The experimental period lasted 22 days, with 17 days for adaptation to the facilities and diets, and 5 days for data collection, replicated in four periods following the methodology adapted from Tomich and Sampaio [28].

The diet consisted of Buffel grass (*Cenchrus ciliares*) hay, Tifton 85 (*Cynodon* spp.), ground corn, soybean meal, mineral mixture and amino acid bypass. The diets were formulated under a 60:40 ratio of roughage-to-concentrate and met the nutritional needs for maintenance according to the NRC [2]. The diet was 89.8% dry matter (DM), 13.2% crude protein (CP), 37.6% non-fiber carbohydrates (NFCs), 51.32% neutral detergent fiber (NDF), 2.38% ether extract (EE) and 70% total digestible nutrients. Animals were fed the diet as TMR ad libitum twice a day, at 8:00 and 15:00 h. Water was easily accessible and at will.

The in situ test was conducted in nonwoven textile (NWT) bags with a porosity of 100 μ and measuring 5 × 5 cm, which were heat-sealed in a sealing machine. Each bag contained approximately 3.0 g of sample containing the microencapsulated material. The bags were incubated incrementally (times of 0, 30, 60, 120, 180, 1440 and 2880 min) and in stages, so that the same treatment was removed from each wax, one at a time to reduce interference during manipulation in the rumen environment. Incubation was carried out for the eight treatments in triplicate, and seven times. After removal, the bags were immersed in cold water until the residue disappeared and to stop degradation by bacteria. They were then washed in running water and placed in a forced-air ventilation oven at 55 °C for 72 h and then weighed on an analytical scale. Immediately after collection at each time point, the liquid was filtered through gauze and the pH was measured using a digital pH meter (Prolab, São Paulo, Brazil), while the temperature was measured using a portable digital thermometer.

### 2.7. Chemical Composition of Diets, Ingredients, and Rumen Liquid

The samples of ingredients and leftovers were pre-dried in a forced-air ventilation oven at 55 °C for 72 h, and were then ground in a Wiley knife-mill with a 1.0 mm sieve. The samples were stored in plastic jars with lids, labeled and subjected to analysis (triplicate) to determine the contents of dry matter (DM; method 967.03), ash (method 942.05), crude protein (CP; method 981.10) and ether extract (EE; method 920.29) according to the Official Method of Analysis Association of Analytical Chemists [29] as presented in Table 1.

### 2.8. Design and Statistical Analysis

The experiment was completely randomized in a 2 × 4 factorial arrangement for Lys protection (core), with two types of encapsulants (shell), beeswax (BW) or carnauba wax (CW), for Lys (encapsulated) amino acid. A randomized block design (RBL) was used to assess degradability in DaisyII ANKOM, with treatments in 2 × 4 factorial arrangements (wax and tannin) as repeated measures over time. Two artificial rumens were used as replicates and four independent jars were used as replicates of the rumen environment inside each apparatus. The evaluation was carried out using a split-plot design, with the treatments in 2 × 4 factorial arrangements (type of wax and level of tannin), replicated in four periods from three sheep and using the same times as the Daisy evaluation. 

The results obtained were subjected to variance analysis in a completely randomized manner and a factorial arrangement with subdivided plots, considering the eight treatments in the plots and the observation times (0, 30, 60, 120, 180, 1440 and 2880 min) of each animal in the subplots (3).

The mathematical model used for analysis of variance is described in Equation (3), as shown below:Yij = µ + A_i_ + M_j_ + P_k_ + H_l_ + L_kj_ + E_ijkl_,(3)
where Y_ij_ is the variable observed in the animal _i_ and material (amino acid bypass) _j_; μ is the general mean; A_i_ is the effect of the animal _i_; M_j_ is the effect of the material _j_; P_k_ is the effect of the period _k_; H_l_ is the effect of time _l_; L_jk_ is the action of period _k_ on diet _j_; E_ijkl_ is the random error.

The data were evaluated through analysis of variance and regression (tannin level), using the software SAS (Statistic Analysis System 9.1) [30], and the means were compared using Tukey’s test at 5% probability.

## 3. Results

The evaluation of the material showed that the microencapsulation yield improved with the inclusion of tannin, ranging from 75 to 79% for bypass Lys microencapsulated in BWLys_0%_ and BWLys_3%_, and from 80 to 84.5% for bypass Lys microencapsulated in CWLys_0%_ and CWLys_3%_, respectively (Figure 1). The microencapsulation efficiency also improved with the addition of tannin, ranging from 81 to 85% for BWLys_0%_ and BWLys_3%_ and from 79 to 89% for CWLys_0%_ and CWLys_3%_. It was observed that the microencapsulation efficiency was high, being above 80%.

Differential exploratory calorimetry showed that thermal events below 100 °C corresponded to the loss of moisture in each material. BW showed an endothermic event (event 1) at 68 °C, related to its melting, and another endothermic event (event 2) at 385 °C, related to its thermal degradation (Figure 2a), while Lys showed three endothermic events at 83, 264 and 309 °C, and tannin showed an endothermic event at 99 °C. 

As for the microencapsulated systems (bypass Lys), it was observed that the events of the isolated phases (wax, Lys and tannin) remained the same or changed according to the presence, absence or increasing level of tannin. The microencapsulated Lys showed a higher thermal degradation temperature (event 3 for Lys and event 4 for microencapsulated materials) than its free form, attesting to the protection of the BW encapsulating matrix.

The CW showed an endothermic event at 82 °C, related to its melting point (Figure 2b), since it melts between 80 and 85 °C. It was also observed that microencapsulated Lys showed a higher thermal degradation temperature (Event 3) than its free form, attesting to the protection of the CW as an encapsulating matrix. This protection against temperature was close for CWLys_0%_ and CWLys_1%_, with a higher value for CWLys_2%_ and a lower value for CWLys_3%_. CWLys_2%_ showed the first endothermic event at 76 °C, which may be the result of the overlap between the melting point temperatures of CW and the event presented by the tannin–lys interaction or even a material with more moisture.

There was no interaction effect between the level of tannin and the type of wax on water activity (WA; Figure 3A), and the DM (Figure 3B) and CP (Figure 3C) contents. However, there was an effect of the wax; for the treatment with 1% inclusion of tannin, CW showed higher WA (*p* < 0.05) compared to BW, and for the inclusion of 3% tannin, the behavior was the opposite with BW, showing higher WA (*p* < 0.05). Among the tannin levels, BWLys_1%_ showed lower WA when compared to the BWLys_0%_ and BWLys_3%_ treatments. For CW, the addition of tannin, regardless of the level, reduced WA in comparison to the treatment without tannin inclusion (CWLys_0%_). 

There was an effect of type of wax and tannin levels (*p* < 0.05) on the DM content of the materials (Figure 3B). CW had a higher DM content compared to BW for the 1% and no-tannin inclusion levels (*p* <0.05), with no difference (*p* > 0.05) between the 2 and 3% tannin inclusion levels. BWLys_2%_ and BWLys_3%_ showed higher DM contents (*p* < 0.05) compared to the other levels, with the control treatment (BWLys_0%_) showing the lowest DM content (93.41%). However, for CW, the 1, 2 and 3% tannin levels were similar (*p* > 0.05) with a higher DM content than the treatment without added tannin BWLys_0%_ (94.75%). When the waxes are compared, levels 2 and 3% are similar for each wax. For the control levels 0 and 1% tannin, CW stands out with higher values than BW.

For both BW and CW Lys protection, the CP concentration increased as the tannin level increased, varying from 20.94% to 22% between the treatments with no added tannin (BWLys_0%_) and 3% added tannin (BWLys_3%_). For protection with CW, the increase was 20.44 and 23.06 between CWLys_0%_ and CWLys_3%_ systems. There was a statistically significant difference between the levels of tannin (*p* < 0.05) in which the 2 and 3% inclusion levels were similar and higher than the other levels. When comparing the waxes, no differences were observed in the treatments without tannin. However, for the other levels of tannin inclusion (1%, 2%, and 3%), CW showed higher (*p* < 0.05) CP levels compared to BW.

All the components used in the microencapsulation as well as the bypass Lys were subjected to scanning electron microscopy (SEM). Image 4A showed an irregular surface with several protuberances and depressions on CW, indicating a complex and porous texture, which is intact, showing difficult invasion of its interior (Figure 4A). 

The surface of CW is highly textured with irregular shapes and there are structures close to flakes or plates scattered across the surface, which may indicate the presence of wax crystals. The BW (Figure 4B), on the other hand, has a very uneven and rough surface. There are several elevations and depressions, as well as formations that resemble fragments or flakes. Some of these fragments seem to be superimposed on each other, which contributes to the complex texture. Its porosity is clearly visible in the image. There are many small pores scattered across the surface. These pores vary in size and shape, which may indicate the complexity of BW’s internal structure. Its appearance is that of a complex and detailed structure.

In Figure 4C, Lys shows a rough surface texture with various elevations and depressions, with some areas appearing smoother. There are no distinct or recognizable shapes on its structures. As for the tannic extract of *Mimosa tenuiflora* (Figure 4D), the image showed an irregular and textured surface; crystalline structures or mineral compounds seem to be present, with irregular shapes and varying sizes. The image shows a rough and uneven surface, with various shapes and textures visible, next to a rocky terrain. 

Figure 4E, in which the carnauba wax is encapsulating the Lys, shows particles with a highly textured surface with irregular shape, indicative of a porous structure, so it may indicate that the particles have a large surface area, which could be useful in applications such as catalysis or adsorption. Figure 4F, in which beeswax encapsulates Lys, shows a highly magnified texture that resembles a rough, uneven surface with several clusters of materials. Figure 4G (BW + Lys + tannin) shows a highly magnified texture of the surface of this material composed of an irregular surface with agglomerates of various sizes that create a rough texture. The CW + Lys + tannin (Figure 4H), on the other hand, shows the highly enlarged texture of a rough and scaly sample with several layers and protuberances.

There was no effect of interaction between wax type and tannin level (*p* > 0.05) on in vitro DM degradation and retention (Figure 5). Regardless of the type of wax, those encapsulated with 3% tannin showed lower retention (65.40%) and higher degradation (34.60), differing from the values observed for those encapsulated with 1 and 2% tannin added to the encapsulating material. The encapsulates without the tannin association showed intermediate retention and degradation, being close (*p* > 0.05) to the other encapsulates.

In vitro rumen degradation of DM for both waxes increased over time (Figure 5). BWLys degradation started at 4.18% and was slower than CWLys, reaching 37.13% at 2880 min. CWLys degradation started at 6.89% in the first hour and gradually increased until it reached 51.86% DM degradation at 2880 min. At zero incubation time (0 min), the degradation rates of DM for BWLys and CWLys were similar (*p* > 0.05), and the greatest differences occurred at the final times (1440 and 2880 min) with the encapsulates using CWLys showing higher values (*p* < 0.05) of degradation (51.86%) at the final time when compared to BWLys (37.13%). It was found that at zero incubation time (0 min), the retention rates (Figure 5) of the DM for CWLys and BWLys were 93.10 and 95.86%, respectively, with no statistical difference (*p* > 0.05). With the increasing incubation time, DM retention decreased for both waxes. At the end of the incubation period (2880 min), the retention of DM for CWLys (48.13%) was lower (*p* < 0.05) than BWLys (62.86%). 

The BW showed higher retention and lower in situ degradation of DM in comparison to CW (Figure 6a,b). The inclusion of 1 and 2% tannin increased retention for both encapsulated materials (BW and CW), being higher for BW. 

Both the N and the CP content of the encapsulates as a function of the tannin level, regardless of the time they were incubated in situ, showed an increasing trend in CP when the tannin level was raised, so it can be observed that the tannin level influences the increase in CP repeatedly (Figure 7a). In addition, CP showed an upward trend as the level of tannin increases, indicating a positive correlation between the level of tannin and CP. Over the incubation time, there was a decrease in N and CP levels (Figure 7b). Both curves show a significant decrease in the first 180 min and stabilization after that. 

Rumen pH (Figure 8a) and temperature (Figure 8b) were similar for the animals fed regardless of the wax (BW *vs* CW) used and the level of tannin (*p* < 0.05). For the two waxes with 0% tannin, the maximum temperature was 39.3 °C and the minimum was 38.5 °C at the start of incubation (0 min). With 3% tannin, the maximum temperature was 39 °C and the minimum was 38.5 °C at the last incubation time (2880 min) for both waxes. For BW and CW without the inclusion (0%) of tannin, the maximum pH value was 7.19 and the minimum was 6.03. Both waxes with the addition of 3%TJP showed values of 6.83 (max) and 5.09 (min). The rumen pH remained constant throughout the incubation periods.

## 4. Discussion

The results obtained for the microencapsulation yield of Lys with BW (BWLys) or CW (CWLys) associated with tannin as a wall material indicate that the emulsification and oven-drying technique was adequate, with no significant losses during the processing of the materials, showing microencapsulation yields of over 80% in all concentrations. 

Oven drying at a higher temperature can form a layer of crust around the particles, which can result in less mass transfer. On the other hand, freeze-drying involves the sublimation of ice under low pressure, so it helps to remove bound water through the phenomenon of substance removal (desorption). According to Carvalho Neto et al. [14], these losses can be attributed to the intrinsic viscosity of the sample and its tendency to solidify at room temperature. This is supported by Etchepare et al. [31] who recommend considering these factors in any encapsulation production process.

The results obtained validate the effectiveness of the emulsification technique for the microencapsulation of the components (core and encapsulant) investigated in this study. This indicates that emulsification is a suitable technique for the microencapsulation of these specific components, corroborating the robustness and applicability of the method in question. Our results for CW were lower compared to the data presented by Carvalho et al. [13], who worked with microencapsulated urea through emulsification/lyophilization, and found values of 92.53% without sulfur and 90.80% with sulfur for the encapsulation yield. In this case, the higher yield found by Carvalho Neto et al. [13] was due to the method of drying the material under low pressure, which helps to remove the bound water through desorption.

Da Silva Aguiar et al. [12] used the emulsification/lyophilization technique for the microencapsulation of urea enriched with a sulfur source, using CW and BW as wall materials, and obtained values ranging from 92.9% to 107%. In contrast, Carvalho Neto et al. [14] used the same technique and obtained yields for CW in the methionine microencapsulation process of 82.33% and 78.5% for the 2:1 and 4:1 formulations, respectively. Although the technique used by the authors to encapsulate methionine is different, i.e., by fusion–emulsification and oven-drying, the encapsulation yields were close.

In general, amino acids can have a consistent impact on yield, regardless of the technique used, which reinforces the claim of Martin et al. [32] that the technique and the nucleus play significant roles in the efficiency of the process. However, when the microencapsulation technique changes, but the substance to be encapsulated is an amino acid, the results tend to be close.

The results obtained indicate that both waxes showed high efficiency in the microencapsulation process, with BW (>98%) being more efficient than CW (±95%). Microencapsulation efficiency is an important parameter in assessing the quality and effectiveness of nutritional formulations. Through careful analysis, it is possible to optimize this process to improve stability, protection and nutrient release [12]. Compared to the study of Carvalho Neto et al. [14], who used CW to microencapsulate methionine in two systems (2:1 and 4:1) using the fusion-emulsification technique, the microencapsulation efficiency results of the present study were lower. In their study, both systems achieved a high microencapsulation efficiency of over 97.5%. Notably, the 2:1 system achieved an MIE of 99.65%.

The graphical representations of differential exploratory calorimetry indicate that the melting temperature of the waxes in microencapsulation systems remained constant compared to the unencapsulated waxes. This attests to the chemical compatibility between the Lys and the encapsulation materials, ensuring protection without interactions which, if they occurred, could affect the release of the Lys. Maintaining the characteristic melting point of the waxes also indicates that it is safe to store the microparticles or use them in processing involving heat up to around 68 °C for the systems with BWLys, and 82 °C for the systems with CWLys.

The differential scanning calorimetry technique also makes it possible to determine the melting temperature (endothermic events) of the materials, a factor of great influence in a controlled release system in which the encapsulant is a wax, providing information related to its stability and the appropriate storage conditions for the material, as well as the degree of purity of the materials used in the formulations. The decrease in calorimetry for the higher level of tannin inclusion is explained by the fact that the more material inserted into the microparticle, the greater the possibility of more core (Lys) close to the surface, i.e., more susceptible to degradation or faster release [33]. This is because the differential exploratory calorimetry technique does not depend on changes in mass, but rather on the energy present in the materials, which can be modified through chemical processes such as oxidation [34] and reduction [11]. 

In general, although CW showed greater thermal stability, a higher melting point and greater hardness than BW, both offered thermal protection to Lys compared to its free form, and this protection can be reproduced in the rumen environment [34], enabling gradual release [11] and optimizing the use of this ingredient in ruminant diets. CWLys had a melting point of 82 °C, which is close to the ranges observed by Milanovic et al. [35] and Lacerda et al. [36], ranging from 80 to 86 °C. On the other hand, the melting point of BWLys was lower (68 °C), corroborating the results obtained by Penagos et al. [37] who obtained 64 °C and by Carvalho et al. [13] who used the same methodology and observed a melting point of 68 °C.

The water activity (WA) of bypass Lys showed that all the formulations were microbiologically safe and stable, with no risk of degradation. The average values obtained in this study for WA were very low, ranging from 0.48 for BW to 0.44 for CW, in line with the recommendation of Gock et al. [38], who established a minimum value of 0.7 to allow microorganisms to germinate. Water content is crucial in food manufacturing, as it directly influences control of the rate of degradation by microorganisms, as well as enzymatic and chemical reactions that occur during storage.

The CP content observed in this study allows us to infer that the addition of tannin from *Mimosa tenuiflora* hay not only increased the encapsulation efficiency but also increased the CP content proportionally to the addition of tannin. Possibly, the tannin addition formed complexes for the protein that allowed the protection of proteins against degradation in the rumen, as described by Da Silva Aguiar et al. [12] when they evaluated digestibility, nitrogen, water, and energy balance in sheep-fed *Mimosa tenuiflora* hay and Buffel grass hay. The difference in results between BW and CW, with a higher CP content for CWLys_3%_, suggests that the type of wax used to encapsulate the Lys may also affect the CP content. This opens up new possibilities for optimizing animal nutrition by manipulating the type of wax and the level of tannin. However, more research is needed to confirm these results and explore their underlying mechanisms.

Regarding the structure of the material through micrographic analysis, CW material presented an irregular surface with several protuberances and depressions, indicating a complex and porous texture that was intact, pointing to the difficult invasion of its interior. This information corroborates the work of Galus et al. [39] and Carvalho Neto et al. [14] who reported that scanning electron microscopy of CW showed a smooth surface, with small elevations and some wrinkles, although sealed, without porosities and intact, indicating the difficult invasion of its interior. Beeswax, on the other hand, had a very irregular, rough surface and several elevations and depressions, as well as formations resembling fragments or flakes. Unlike Carvalho et al. [13], in our study, BW had a slightly irregular, intact, sealed, non-porous and smooth surface.

Encapsulation with CW showed greater degradation of DM in comparison to BW at all incubation times. This can be explained by the difference in the chemical composition of the two waxes, with CW containing more fatty acid esters, which are more susceptible to hydrolysis than the hydrocarbons present in BW [40].

The DM degradation of BW and CW was influenced by the level of tannin added, with the 3% tannin level showing lower degradation and higher DM retention than the encapsulated BW without tannin. This proves the effect of tannin in forming insoluble complexes with the proteins present in the waxes, reducing their availability for rumen degradation [20,22], thus allowing, with the addition of tannin, a higher rate of protection of the Lys and causing less release of the material during the process of attack by microorganisms. Condensed tannins have an additional reaction mechanism because they are more resistant to removal by breaking hydrogen bonds [41]. It has been suggested that this additional interaction is the covalent reaction between the protein and the aromatic carbon of the tannin molecules through quinoidal structures [42]. 

Therefore, when the encapsulates retain more CP and DM, the animals can make better use of these compounds. This can lead to less loss of material and therefore a higher passage rate. By increasing the retention of CP and DM in encapsulates, nutrient passage and utilization at the small intestine level can be improved. This is because the balance of methionine and Lys leads to greater efficiency in protein metabolism [43] and, consequently, reduces nitrogen losses [44]. According to Araújo et al. [45], when Lys reaches the rumen, it is rapidly degraded and does not reach the small intestine in sufficient quantities, making it difficult to meet metabolizable protein requirements. Carvalho Neto et al. [14] highlight the need for amino acids in the diet for protein synthesis activities in various tissues, as well as for microbial efficiency to maintain the proper functioning of the rumen environment.

The animals had an average rumen fluid temperature during incubation of the materials of 38.8 °C, with no influence from the type of wax or the addition of tannin to the encapsulating material. According to Berchielli et al. [46] and Millen et al. [47], the ideal rumen temperature should be around 38 to 42 °C. The average pH observed for rumen fluid was 6.28, also without any influence from the incubation of the materials. Several studies state that a favorable environment for bacterial proliferation occurs with rumen pH ranging from 6 to 7.2 [47,48]. All materials maintained ruminal pH and temperature values in sheep within the normal range, which is fundamental for the health and digestive efficiency of ruminant animals. Maintaining this balance is essential to ensure that microorganisms perform their function properly and that animals make the most of the nutrients in their diet.

## 5. Conclusions

The use of BW and CW matrices as encapsulating agents, using tannin as an adjuvant, was efficient in microencapsulating lysine for passage through the rumen, showing high thermal stability and yields. Formulations with 3% tannin, regardless of the type of wax used, resulted in a higher content of DM and CP retained in the microcapsules. 

The encapsulation of lysine in both waxes led to an increase in DM and undegraded CP without compromising rumen pH and temperature. Carnauba wax proved to be more efficient than BW in protecting CP from rumen degradation. Therefore, the recommended microencapsulated system is CWLys_3%_.

## 6. Patents

Two patents were filed resulting from the work reported in this manuscript (registration number BR10202301738; and registration number BR102023017).

## Figures and Tables

**Figure 1 animals-14-02895-f001:**
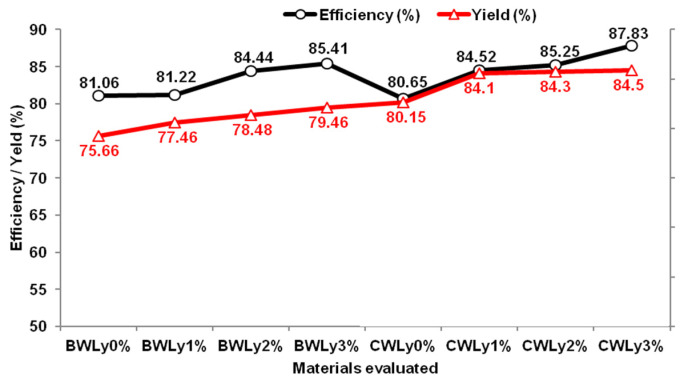
Efficiency and yield of lysine bypass encapsulated with beeswax (BW) and carnauba wax (CW) with tannin levels of 0, 1, 2 and 3% produced by the fusion–emulsification method.

**Figure 2 animals-14-02895-f002:**
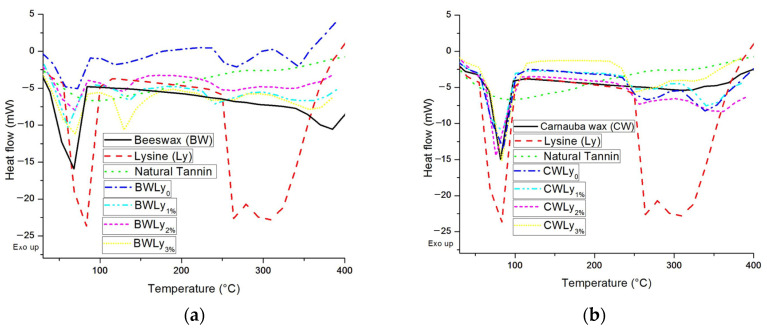
Differential scanning calorimetry (DSC) curve of lysine, beeswax (**a**) and carnauba wax (**b**) and lysin bypass (BWLys and CWLys) with natural tannin levels of 0, 1, 2 and 3% produced by the fusion–emulsification method.

**Figure 3 animals-14-02895-f003:**
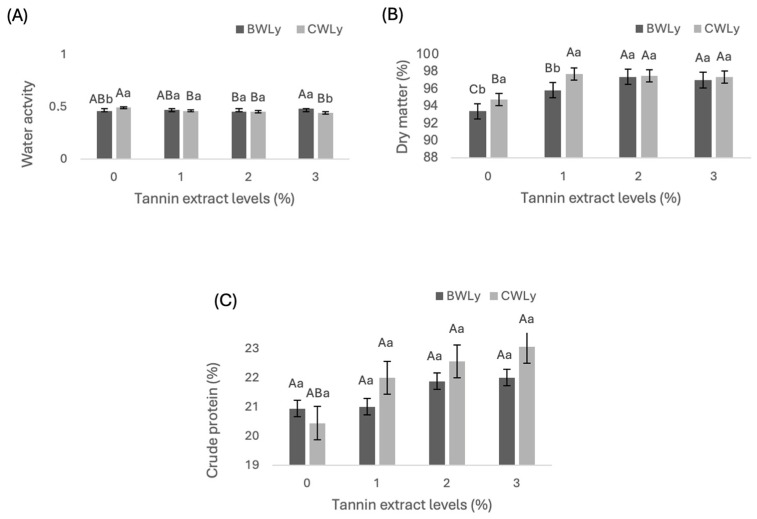
(**A**) Water activity (WA), (**B**) dry matter (DM), and (**C**) crude protein (CP) of lysin bypass encapsulated into beeswax and carnauba wax (BWLys and CWLys) with natural tannin levels of 0, 1, 2 and 3% produced by the fusion–emulsification method. Different letters indicate significant differences at *p* < 0.05; same letters indicated no significant differences (*p* > 0.05).

**Figure 4 animals-14-02895-f004:**
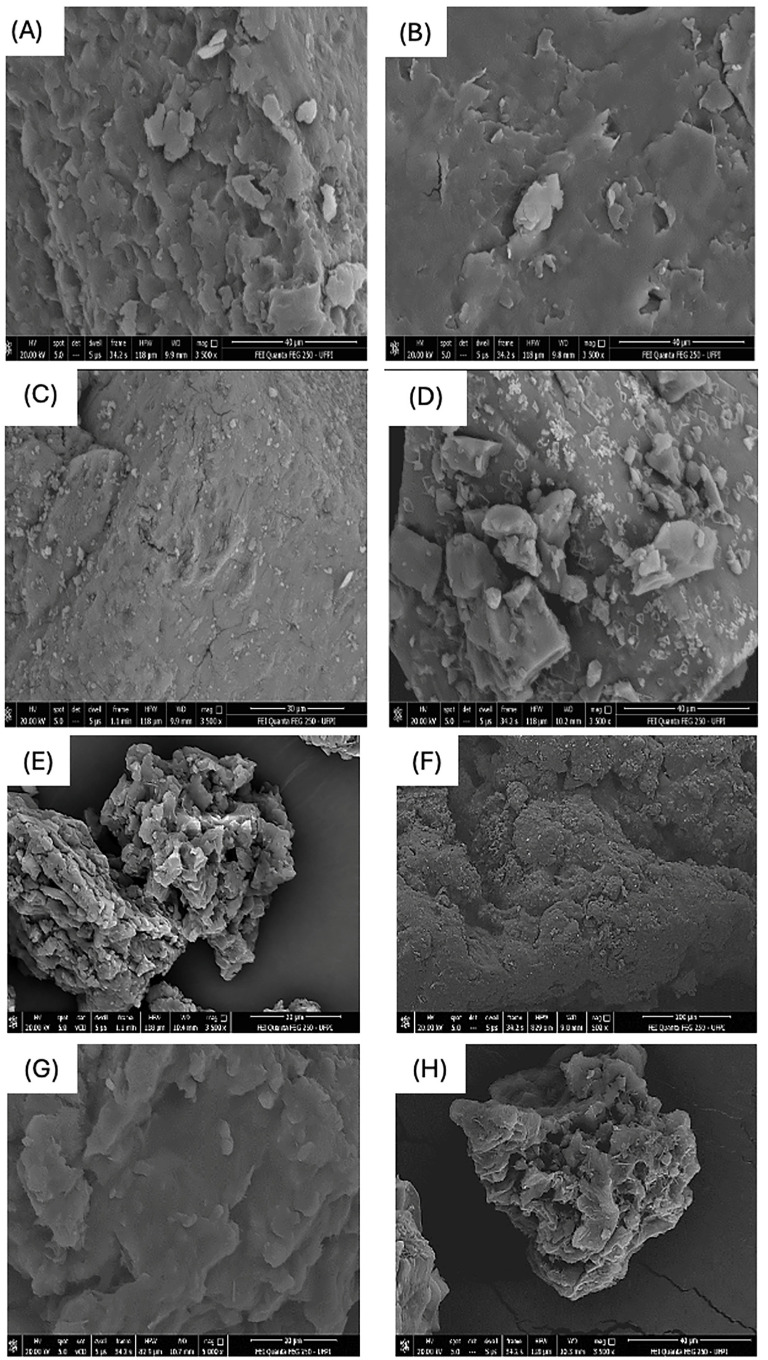
Scanning electron micrographs: (**A**) carnauba wax; (**B**) beeswax; (**C**) lysine; (**D**) *Mimosa tenuiflora* tannic extract; (**E**) carnauba wax + lysine; (**F**) beeswax + lysine; (**G**) beeswax + lysine + *Mimosa tenuiflora* tannic extract; (**H**) carnauba wax + lysine + *Mimosa tenuiflora* tannic extract.

**Figure 5 animals-14-02895-f005:**
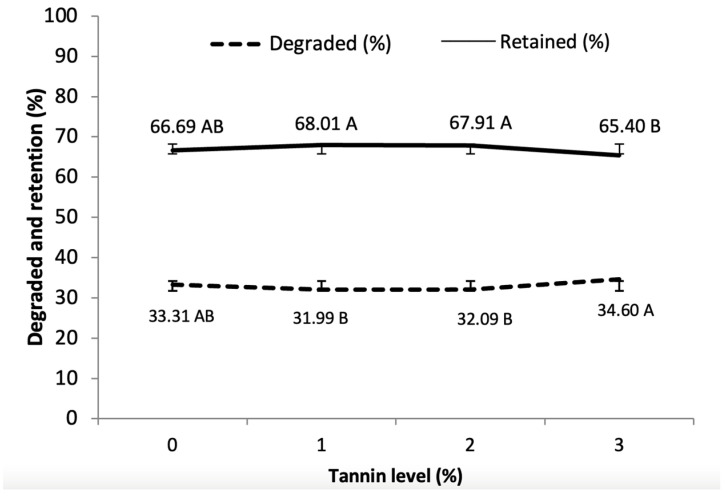
Degradation and retention of dry matter (DM) of encapsulates as a function of the level of tannin, regardless of the type of wax and incubation time in the DaisyII ANKOM. Different letters indicate significant differences at *p* < 0.05; same letters indicated no significant differences (*p* > 0.05).

**Figure 6 animals-14-02895-f006:**
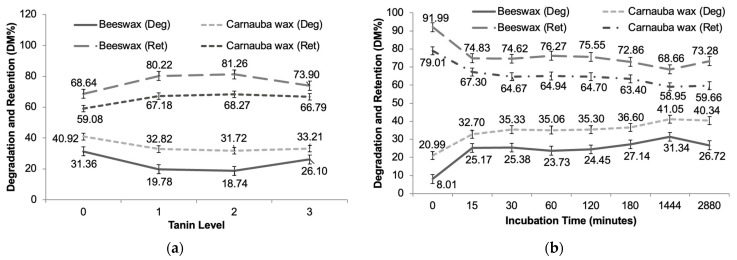
Degradation and retention of dry matter (DM) of lysin bypass encapsulated products as a function of tannin levels (**a**), and in situ incubation time (**b**).

**Figure 7 animals-14-02895-f007:**
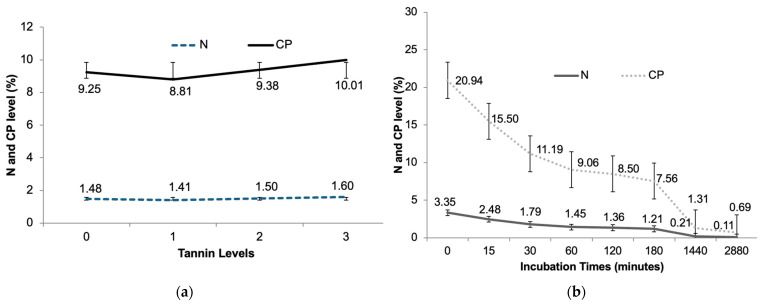
Average nitrogen (N) and crude protein (CP) of lysin bypass encapsulated products as a function of tannin level (**a**) and in situ incubation time (**b**).

**Figure 8 animals-14-02895-f008:**
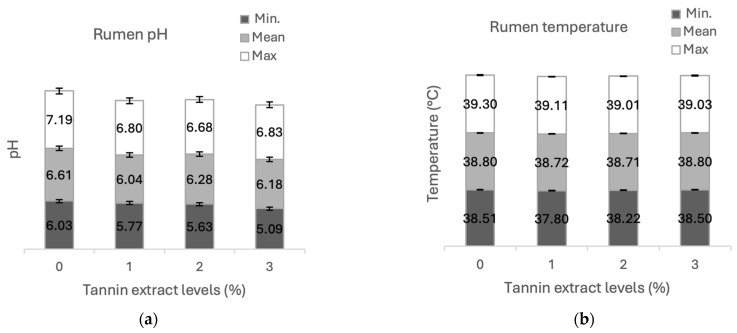
Average rumen (**a**) pH and (**b**) temperature (°C) and at different incubation times in the fistulated animal, independent on the type of wax (beeswax and carnauba) as a function of the level (0; 1; 2 and 3%) of tannin added in material as an adjuvant.

**Table 1 animals-14-02895-t001:** Composition of ingredients used in the production of encapsulated products.

Materials	Chemical Composition (%DM)				
	Dry Matter	Organic Matter	Crude Ash	Nitrogen	Curde Protein
Beeswax (BW)	95.74	99.00	1.00	0.44	2.75
Lysine (Lys)	95.61	98.95	1.05	12.57	78.54
Tannin	88.35	92.42	-	-	-
BWLys_0%_	93.41	98.50	1.50	3.32	20.73
BWLys_1%_	95.86	97.99	2.01	3.32	20.72
BWLys_2%_	97.41	97.97	2.03	3.56	22.23
BWLys_3%_	97.05	97.95	2.05	3.59	22.46
Carnauba wax (CW)	97.98	98.99	1.01	0.45	2.83
Lysine (Lys)	95.61	98.95	1.05	12.57	78.54
Tannin	88.35	92.42	-	-	-
CWLys_0%_	94.75	98.48	1.52	3.21	20.06
CWLys_1%_	97.71	97.97	2.03	3.07	19.17
CWLys_2%_	97.53	97.97	2.03	3.33	20.81
CWLys_3%_	97.39	97.94	2.06	3.39	21.20

## Data Availability

The data presented in this study are available upon request to the author for correspondence.

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
