# Peer review of "New Technology of Rumen-Protected Bypass Lysine Encapsulated in Lipid Matrix of Beeswax and Carnauba Wax and Natural Tannin Blended for Ruminant Diets"

_animals, 2024, doi:10.3390/ani14192895_

Round 1

Reviewer 1 Report

Comments and Suggestions for Authors

See attached pdf.

Comments on the Quality of English Language

English is very good. Only minor grammar to fix. 

Author Response

Overall Comments: I was very impressed by this manuscript and the research done. I believe this is excellent work and a great study investigating different amino acid encapsulation techniques for use in ruminant diets. I have very few comments about the manuscript, and all are associated with grammar and run on sentences.

Response = Thanks. We appreciate the attention, time and reviewers' contributions to the improvement of this version of the manuscript. We have modified the manuscript according to the suggestions and below we present point-by-point answers to the questions. All corrections were addressed, as shown below and in the attached file. Answers to the questions are provided below. All the manuscript changes have been highlighted in red font. 

L81-86: Please rewrite sentences for more clarity. The first sentence is long and confusing and

the second is short.

Response = Thanks. This section was rewritten: “The microencapsulation technique can protect amino acids in the diet of ruminants and allows controlled release at specific sites in the gastrointestinal tract [14]. To allow efficient transport of the nucleus, methionine, or lysine amino acids, the protective substance must protect the compound from attacks by the microbiota in the animal's rumen and gradually release this active substance into the small intestine [15].”

L90: Need to define BW and CW at first use in the main text. You defined them in the simple

summary and abstract but should do the same for main body of manuscript

Response = Thanks. We have defined BW and CW abbreviations at first use in the main text.

L115-121: Entire paragraph is one single run on sentence. Please divide it into two or three

sentences. Confusing as written.

Response = Thanks. This section was rewritten.

L150: Please don’t use abbreviations at the beginning of a sentence. Spell out Beeswax.

Response =Thanks. We have corrected the abbreviations use  at the beginning of a sentence.

L141 and 184: Abbreviate lysine at first use in the manuscript and update all places where it would use the abbreviation.

Response = Thanks. We have abbreviated lysine to “Lys” at first use in the manuscript and update all places where it would use an abbreviation.

L211-218: “The diet consisted of Buffel grass (Cenchrus ciliares) hay, Tifton 85 (Cynodon spp.), corn, soybean meal, mineral salt and encapsulated material. The diets were formulated under a 60:40 ratio of roughage-to-concentrate and met the nutritional needs for maintenance according to the NRC. The diet was 89.8% dry matter (DM), 13.2% crude protein…….”

Response = This section was rewritten. Thank you.

L218-219: add manufacturer of a TNT bag or describe more clearly what a TNT bag is.

Response =Thanks, we have described more clearly what a TNT bags “… nonwoven textile (NWT) bags with porosity of 100 μ and measuring 5 × 5 cm”….

L244: Please add the lysine treatments for the 2x4 factorial in this sentence.

Response = Thanks. We have added the lysine treatments for the 2 x 4 factorial in this sentence.

L307: There was an effect of….

Response = Thanks. This was corrected.

L487: allows us to infer that…

Response =Thanks. Changed as requested.

L487-492: Break this into two sentences. It will make more sense

Response = Thanks. We have broken this into two sentences.

L500-501: End sentence at “interior” and start new sentence talking about Calus and Carvalho Neto papers

Response =Thanks. Changed as requested.

L504: Spell out beeswax at beginning of sentence

Response =Thanks. Changed as requested.

L530: highlights

Response =Thanks. Changed as requested.

L550: Please don’t use abbreviations at the beginning of sentences. Spell out carnauba wax.

Response =Thanks. Changed as requested.

Reviewer 2 Report

Comments and Suggestions for Authors

The authors noted the importance and expediency of studying new ways to protect feed additives based on amino acids in ruminant diets. In my opinion, this is an actual and valuable additive for agriculture. Because it will help to replace more efficient and cheaper feeds with alternatives from a more profitable and economical side, which will allow you to get functional food products. The presented manuscript is relevant because it is devoted to the study of new ways to protect amino acids and their passage through the scar and into the intestines of ruminants. In the future, it will be possible to produce protected amino acids with a different degree of protection (except for fatty and synthetic) shells, which in the future will probably be more economical than current technologies. The aim of the study was to develop and apply a new encapsulated microstructure of a lysine-based bypass protein as a core using an emulsification method and a lipid matrix made of carnauba and beeswax as a shell, enrich it with natural tannins, characterize the material and evaluate its inclusion in the diet of ruminants. In general, the work is relevant and of great interest for further development. However, during the review process, some inaccuracies and recommendations were noted to the authors, to which we would like to receive a more detailed answer: 1. Shortening of table names is too long. 2. The research methodology meets the requirements. 3. The results of the study meet the requirements, and the conclusions are supported by the results.

Author Response

The authors noted the importance and expediency of studying new ways to protect feed additives based on amino acids in ruminant diets. In my opinion, this is an actual and valuable additive for agriculture. Because it will help to replace more efficient and cheaper feeds with alternatives from a more profitable and economical side, which will allow you to get functional food products. The presented manuscript is relevant because it is devoted to the study of new ways to protect amino acids and their passage through the scar and into the intestines of ruminants. In the future, it will be possible to produce protected amino acids with a different degree of protection (except for fatty and synthetic) shells, which in the future will probably be more economical than current technologies. The aim of the study was to develop and apply a new encapsulated microstructure of a lysine-based bypass protein as a core using an emulsification method and a lipid matrix made of carnauba and beeswax as a shell, enrich it with natural tannins, characterize the material and evaluate its inclusion in the diet of ruminants. In general, the work is relevant and of great interest for further development. However, during the review process, some inaccuracies and recommendations were noted to the authors, to which we would like to receive a more detailed answer:

Response = Thanks. We appreciate the attention, time and reviewers' contributions to the improvement of this version of the manuscript. We have modified the manuscript according to the suggestions and below we present point-by-point answers to the questions. All corrections were addressed, as shown below and in the attached file. Answers to the questions are provided below. All the manuscript changes have been highlighted in red font. 

  1. Shortening of table names is too long.

Response = Thank you. We have shortened the tables titles.

  1. The research methodology meets the requirements.

Response = Thank you.

  1. The results of the study meet the requirements, and the conclusions are supported by the results.

Response = Thank you.